# A small molecule that disrupts *S.* Typhimurium membrane voltage without cell lysis reduces bacterial colonization of mice

**Jamie L. Dombach**[1]☯*, **Joaquin LJ Quintana**[1]☯, **Samual C. Allgood**[1], **Toni A. Nagy**[1]¤, **Daniel L. Gustafson**[2], **Corrella S. Detweiler**[1]*

**1** Department of Molecular, Cellular, and Developmental Biology, University of Colorado Boulder, Boulder, Colorado, United States of America, **2** Department of Clinical Sciences, College of Veterinary Medicine and Biomedical Sciences, Colorado State University, Fort Collins, Colorado, United States of America

☯ These authors contributed equally to this work.
¤ Current address: Department of Biochemistry, University of Colorado Boulder, Boulder, Colorado, United States of America
* jamie.dombach@colorado.edu (JLD); detweile@colorado.edu (CSD)

**Data Availability Statement:** All relevant data are within the manuscript and its Supporting Information files.

## Abstract

As pathogenic bacteria become increasingly resistant to antibiotics, antimicrobials with mechanisms of action distinct from current clinical antibiotics are needed. Gram-negative bacteria pose a particular problem because they defend themselves against chemicals with a minimally permeable outer membrane and with efflux pumps. During infection, innate immune defense molecules increase bacterial vulnerability to chemicals by permeabilizing the outer membrane and occupying efflux pumps. Therefore, screens for compounds that reduce bacterial colonization of mammalian cells have the potential to reveal unexplored therapeutic avenues. Here we describe a new small molecule, D66, that prevents the survival of a human Gram-negative pathogen in macrophages. D66 inhibits bacterial growth under conditions wherein the bacterial outer membrane or efflux pumps are compromised, but not in standard microbiological media. The compound disrupts voltage across the bacterial inner membrane at concentrations that do not permeabilize the inner membrane or lyse cells. Selection for bacterial clones resistant to D66 activity suggested that outer membrane integrity and efflux are the two major bacterial defense mechanisms against this compound. Treatment of mammalian cells with D66 does not permeabilize the mammalian cell membrane but does cause stress, as revealed by hyperpolarization of mitochondrial membranes. Nevertheless, the compound is tolerated in mice and reduces bacterial tissue load. These data suggest that the inner membrane could be a viable target for anti-Gram-negative antimicrobials, and that disruption of bacterial membrane voltage without lysis is sufficient to enable clearance from the host.

## Author summary

As bacterial resistance to existing antibiotics increases and expands, scientists are exploring new approaches to combatting bacterial infections. There is a special need for antibiotics against Gram-negative bacteria, which are difficult to treat and can cause devastating

**Funding:** o The work was supported by the National Institute of Health of the United States of America, AI151979 and AI121365 (CSD). The Drug Discovery and Development Shared Resource is funded by P30CA046934 (University of Colorado Cancer Center Support Grant). The funders had no role in study design, data collection and analysis, decision to publish, or preparation of the manuscript.

**Competing interests:** The authors have declared that no competing interests exist.

infections. One underexplored possible antimicrobial target for Gram-negative bacteria is the bacterial cell membrane, a structure essential for viability. Here we describe a small molecule that inhibits a Gram-negative bacterial infection in host cells and mice. This molecule disturbs, but does not permeabilize, bacterial cell membranes under growth conditions that mimic infection. These data indicate that subtle bacterial membrane damage caused by a small molecule augments host innate immune defenses and enables bacterial killing, suggesting a new approach to antibacterial therapy.

## Introduction

The Gram-negative bacterial cell envelope consists of an outer membrane, a cell wall, and an inner membrane [1,2]. The outer membrane is a formidable barrier to chemicals due to the presence of lipopolysaccharide (LPS) in the outer leaflet. The negative charges on LPS are stabilized by recruited cations (e.g., $Mg^{2+}$, $Ca^{2+}$), creating a coat around the bacterium that excludes many antibiotics effective against Gram positive bacteria [3,4]. Between the outer and inner membranes is a cell wall with gaps sufficient to enable small molecules to permeate [5,6]. However, the inner membrane supports efflux pumps that span the cell envelope. These pumps capture and export toxic compounds from the periplasm and the inner membrane [7,8].

During infection, the bacterial cell envelope is attacked by soluble innate immune defenses in body fluids, including serum and the contents of phagolysosomes. This arsenal includes agents such as the C3b component of complement, the membrane attack complex, proteases, human guanylate binding protein-1 (hGBP1), diverse antimicrobial peptides (AMPs), and lysozyme [4,9–15]. Therefore, bacteria likely become susceptible to small molecules during infection, and these small molecules could reach cellular targets due to progressive degradation of the cell envelope. In support of this idea, antibiotics such as azithromycin [16,17] and clofazimine [18] are inactive against Gram-negative bacteria in standard microbiological media but have potency in whole animals.

We have developed a screening platform based on a tissue culture cell infection assay with the Gram-negative bacterial pathogen *Salmonella enterica* serotype Typhimurium (*S.* Typhimurium). This platform has identified compounds that enable the clearance of bacteria from host cells but have no effect on bacterial growth in standard media. These compounds include clofazimine and JD1. Clofazimine is an anti-mycobacterial agent demonstrated to have activity against *S.* Typhimurium in macrophages and mice but not in standard broth [18]. JD1 is a small molecule (406 g/mol) that rapidly destroys the inner membrane under broth conditions that mimic host defenses by weakening the LPS layer and/or reducing efflux pump activity. Further, JD1 reduces bacterial colonization of macrophages and the murine spleen [19]. These observations suggest that an empirical approach to identifying small molecules that synergize with host innate immunity has the potential to reveal new chemicals with *in vivo* antimicrobial activity. Here we describe D66, a small molecule that enables bacterial clearance from macrophages and mice. Under conditions where the outer membrane or efflux pumps are compromised, D66 disrupts the inner membrane without rapid permeabilization, revealing that modest disturbances of the inner membrane may be sufficient for antimicrobial activity.

## Results

### A small molecule prevents *S.* Typhimurium survival in macrophages

D66 is a hydrophobic small molecule (cLogP of 4.73, 378 g/mol) that contains two aromatic groups, a seven-membered saturated heterocyclic ring with two nitrogens (a 1,4-diazepane),

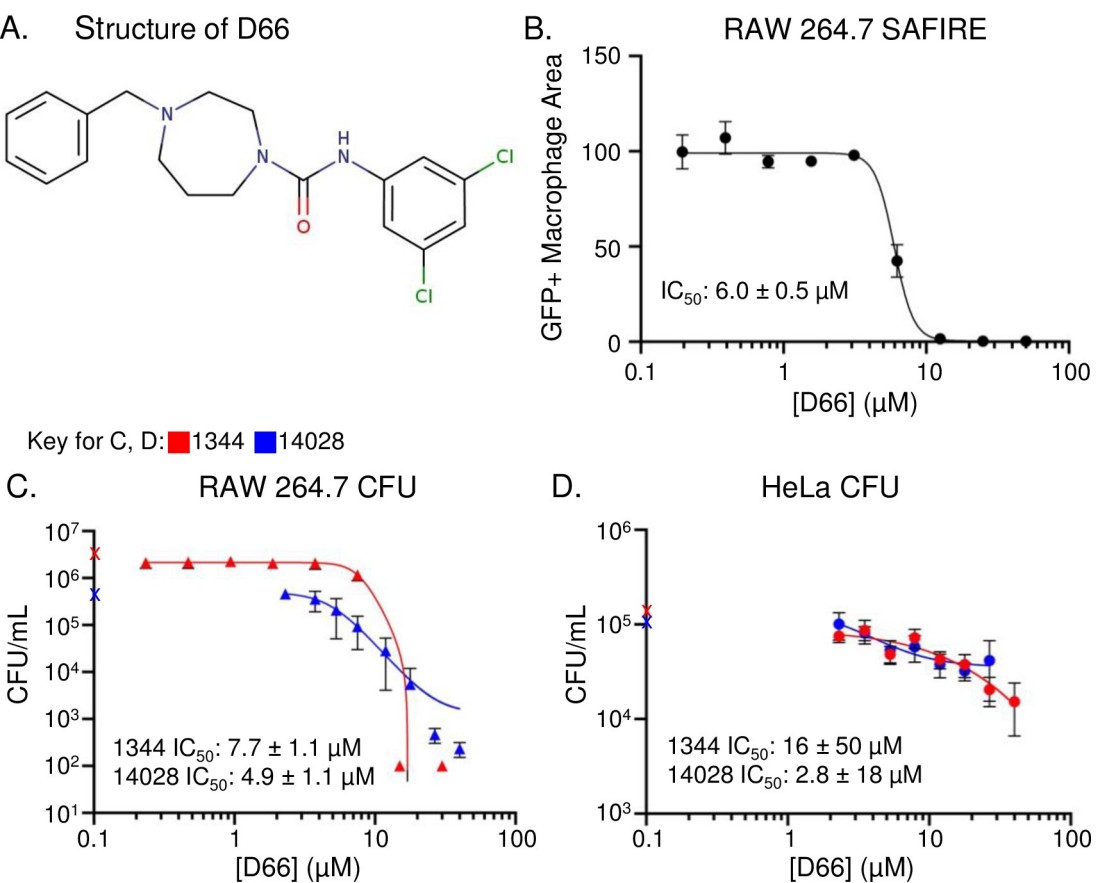

**Fig 1. D66 is a small molecule that prevents *S*. Typhimurium replication and/or survival in macrophages.** A). Structure of D66. B) RAW264.7 macrophage-like cells were infected with *S*. Typhimurium harboring a chromosomal *sifB*::*gfp* reporter, treated with DMSO or D66 at two hours after infection, and monitored for bacterial accumulation (GFP+ Macrophage Area) over 16 hours using the SAFIRE assay. The $IC_{50}$ value is indicated. Mean and SD of biological duplicates with technical duplicates across 9 dilutions of D66. C, D) CFU assays were performed with RAW264.7 or HeLa cells infected with *S*. Typhimurium SL1344 or 14028 for two hours and treated for 16 hours with DMSO or 1.5-fold dilutions of D66, prior to lysis and plating for CFU. Symbols on the Y-axes are the CFU value from DMSO-treated samples. The $IC_{50}$ values are indicated. Mean and SEMs of biological triplicates with technical duplicates across 8 dilutions of D66.

and a urea (Fig 1A). This compound, which has not been previously studied, was found to be highly active in a high content screening platform known as SAFIRE (Screen for Anti-infectives using Fluorescence microscopy of IntracellulaR Enterobacteriaceae) [20]. SAFIRE reports the accumulation of *S*. Typhimurium within macrophages based on *sifB*::gfp expression [21]. Here we found that the half maximum inhibitory concentration ($IC_{50}$) for GFP signal in macrophages was $6.0 \pm 0.5$ μM (Fig 1B). To establish whether D66 reduces bacterial load or interferes with GFP expression, we lysed infected macrophages that had been treated with the compound for 16 hours and plated for bacterial colony forming units (CFU). The CFU $IC_{50}$s of two virulent *S*. Typhimurium strains were $7.7 \pm 1.1$ μM (SL1344) and $4.9 \pm 1.1$ μM (14028) (Fig 1C). Since *S*. Typhimurium can replicate in other cell types, we also tested the ability of D66 to reduce bacterial load in HeLa cells, which are derived from epithelial cells. The compound had little effect; while the calculated $IC_{50}$s are low, the 5-10-fold reduction in CFU, compared to the >1000-fold CFU reduction in macrophages does not give confidence that D66 is highly active in HeLa cells (Fig 1D). These data demonstrate that D66 enables the killing of intracellular *S*. Typhimurium in macrophages.

## D66 inhibits bacterial growth under conditions that compromise the cell envelope

To determine whether D66 could act directly on bacteria, we exposed bacteria to the compound under standard broth conditions, in lysogeny broth (LB) or cation-adjusted Mueller-Hinton Broth (MHB) [22–24] (Table 1 and Fig 2A and 2B). No inhibition of re-growth from stationary phase was observed, consistent with previous compounds identified with the SAFIRE assay [18–20,25]. However, under conditions that compromise the LPS layer of the outer membrane and/or efflux pumps, D66 prevented growth. Specifically, in the presence of the cAMP polymyxin B (PMB), D66 had a calculated minimum inhibitory concentration 95 (cMIC$_{95}$, defined as the concentration at which 95% of growth of the corresponding strain was inhibited) of 54 μM. In these experiments, polymyxin B was at a sublethal concentration [0.5 μg/mL], which we previously showed permeabilizes the S. Typhimurium outer, but not inner, membrane [18]. In contrast to PMB, the polymyxin B nonapeptide (PMBN) did not potentiate D66: concentrations of PMBN [20 μg/mL] that in MHB enable novobiocin to reach cellular targets [26] did not enable D66 to inhibit growth. Both PMB and PMBN bind LPS, but the latter lacks the fatty acid tail and is less disruptive to the outer membrane [27–29]. We also found that strains lacking genes encoding efflux pump subunits (*acrAB* or *tolC*) are sensitive to D66, compared to the parent strains. This effect that was stronger in MHB than in LB (Table 1 and Fig 2B). These data indicate that D66 slowly traverses an intact outer membrane and is captured and expelled by efflux pumps. Outer membrane permeabilization or loss of efflux pump subunits thus facilitate D66 antimicrobial activity. D66 therefore has a direct, negative effect on bacterial growth under conditions that compromise the outer membrane and/or efflux pumps.

## The outer membrane does not appear to be permeabilized by D66

The sensitivity of the *ΔacrAB* and *ΔtolC* mutants to D66 suggests that D66 needs to cross the outer membrane to mediate its effect. However, if D66 were to damage the outer membrane and thereby facilitate its own entry and that of PMB into the cell, this could explain why the compound is potentiated by both PMB and by mutations in efflux pumps. We therefore established whether treatment with D66 potentiates growth inhibition with novobiocin, an antibiotic that cannot traverse the outer membrane [26]. Control compounds included PMB and JD1, which permeabilizes inner membranes and is not expected to potentiate novobiocin

**Table 1. Concentrations of D66 that inhibit S. Typhimurium and E. coli growth.**

| Species (strain) Condition/mutation | IC$_{50}$ [1] in SAFIRE | MIC$_{50}$ [2] in LB | | cMIC$_{95}$ [3] in LB | MIC$_{50}$ in MHB | cMIC$_{95}$ [3] in MHB |
|---|---|---|---|---|---|---|
| | μM | μM | μg/mL | μM | μM | μM |
| S. Typhimurium (SL1344) | 6.0 ± 0.5 [4] | >150 | >57 | >150 | >150 | >150 |
| w/ PMB (0.5 ug/mL) | NA | 45 ± 4 | 17 | 54 | 40 ± 1 | 46 |
| *ΔacrAB* | NA | 128 ± 2 | 48 | 142 | 81 ± 6 | 116 |
| E. coli (K-12) | NA | >150 | >57 | >150 | >150 | >150 |
| *ΔtolC* | NA | 47 ± 1 | 18 | 52 | 36 ± 1 | 43 |

[1] IC$_{50}$: The concentration of D66 that prevents half of the accumulation of GFP signal in macrophages

[2] MIC$_{50}$: The concentration of D66 that prevents half of the growth of the corresponding bacterial strain in broth

[3] cMIC$_{95}$: The calculated concentration of D66 that prevents 95% of growth of the corresponding bacterial strain in broth, derived from the non-linear regression calculated from the MIC curves. These values were used for experiments, as indicated in figure legends.

[4] Fig 1B, with the SL1344 *sifB::gfp* strain

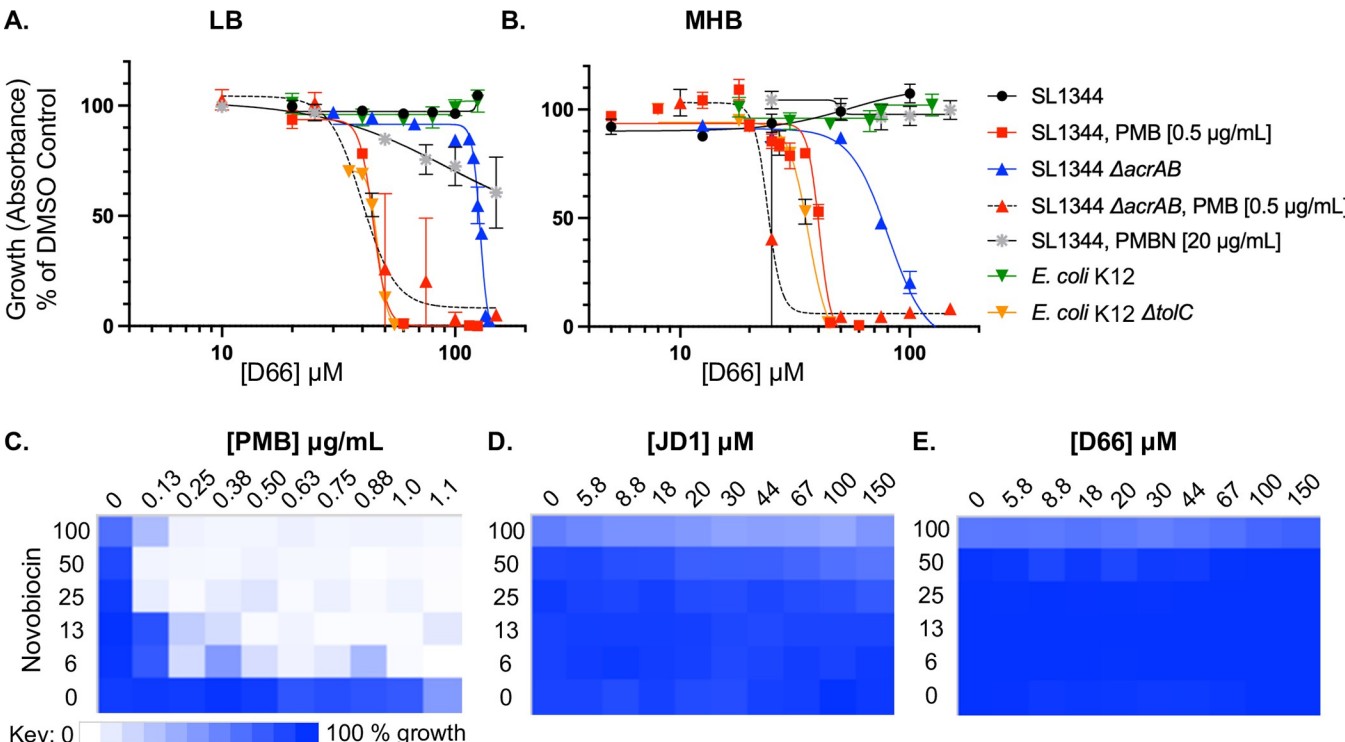

**Fig 2. D66 inhibits bacterial growth under conditions that damage the cell envelope and does not permeabilize the outer membrane.** A, B) Dose response curves monitoring *S.* Typhimurium (SL1344) or *E. coli* (K12) growth from an $OD_{600}$ of 0.01, normalized to growth in 2% DMSO under the indicated condition. Mean and SEM of at least three biological replicates performed with technical triplicates; curve fit: sigmoidal, 4PL. C—E) Checkerboard assays of *S.* Typhimurium SL1344 growth from an $OD_{600}$ of 0.01 in LB for 18 hours with novobiocin (up to 100 µg/mL) and (C) PMB (up to 48 µM), (D) JD1 (up to 150 µM), or (E) D66 (up to 150 µM). Growth was normalized to growth in 2% DMSO, with the darkest blue representing 100% growth and white representing 0%.

[19,30] (Fig 2C and 2D). D66 did not potentiate novobiocin at concentrations up to 150 µM (Fig 2E). These data indicate that D66 does not appear to permeabilize the outer membrane, indicating it inhibits growth by an alternative mechanism(s).

## D66 disrupts voltage without permeabilizing the bacterial inner membrane

Since D66 is hydrophobic (cLogP = 4.73), it could inhibit bacterial growth by affecting the inner membrane. Therefore, we established whether the compound disrupts the proton motive force using the fluorescent probe 3,3'-dipropylthiadicarbocyanine iodide [$DiSC_3(5)$]. $DiSC_3(5)$ accumulates in membranes that have an electrochemical and proton gradient, where its fluorescence is partially quenched [31]. As a control, we monitored $DiSC_3(5)$ fluorescence in the presence of D66 in cell-free medium and noted that D66 quenches $DiSC_3(5)$ signal in a concentration-dependent manner (Fig A in S1 Fig). To enable $DiSC_3(5)$ and D66 to traverse the outer membrane [18], *S.* Typhimurium cells were grown in LB with PMB [0.5 µg/mL] and treated with DMSO, JD1, or D66. As expected, JD1 increased $DiSC_3(5)$ fluorescence [19] (Fig 3A). Once we had controlled for the quenching effect of D66 on $DiSC_3(5)$ fluorescence, it became apparent that D66 exposure rapidly increased the fluorescence of $DiSC_3(5)$ in a dose-dependent manner (Fig 3A). These data indicate that the compound disrupts voltage across the bacterial inner membrane.

At least two activities of D66 could rapidly disrupt voltage, physical membrane perturbation and subsequent permeabilization, or depolarization. To determine whether D66 permeabilizes

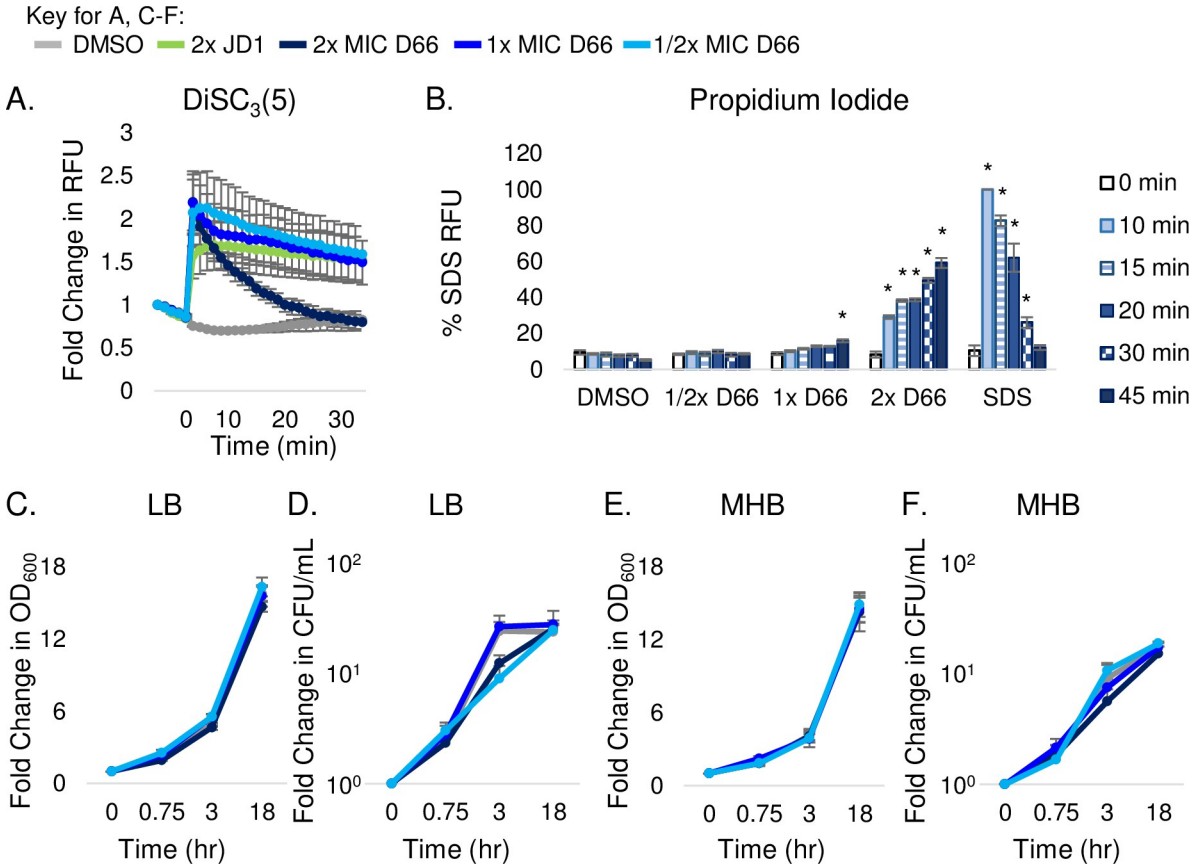

**Fig 3. D66 rapidly perturbs the bacterial cytoplasmic membrane with minor disruption of barrier function.** Mid-log phase *S*. Typhimurium cells grown in LB with 0.5 μg/mL PMB were used for all experiments. D66 MIC$_{95}$ concentrations are provided in Table 1. A) Cell membrane potential was monitored with the fluorescent dye DiSC3(5). Cells were treated at time 0 with DMSO, JD1 [70μM], or D66. Data were normalized to DMSO at time 0 and corrected for the quenching effect of D66 (Fig A in S1 Fig) B) Cell membrane permeability was monitored by PI fluorescence. Cells were treated at time 0 with DMSO, SDS [0.005%] or D66. Samples were processed at the time points shown. A one-way ANOVA with a Dunnet's multiple comparison test, * $P < 0.005$. C-F) Growth curves and kill-curves of cells treated at time 0 with either DMSO or D66. Culture aliquots were monitored for OD$_{600}$ (C, E) or plated for enumeration of CFU (D, F). Data are presented as fold change. Mean and SEM of three biological replicates with technical triplicates are shown in all panels.

the inner membrane under the same outer membrane-permeabilizing conditions (LB with PMB [0.5 μg/mL]), we used the cell impermeant dye propidium iodide (PI), which enters the cell upon inner membrane damage. After 10 minutes of treatment with 2x MIC$_{95}$ D66 or the SDS positive control, PI signal increased, indicating it had crossed the inner membrane and bound DNA (Fig 3B). However, at 1x MIC$_{95}$, 45 minutes elapsed prior to an increase in PI fluorescence. Thus, D66 disrupts membrane voltage immediately even at 1/2x MIC$_{95}$, but higher concentrations and longer incubation periods are required to permeabilize the inner membrane.

Another measure of membrane permeabilization is cell lysis. However, in cells grown under outer-membrane perturbing conditions, treatment with D66 at 2x MIC did not reduce the absorbance nor the CFU of bacteria, and instead the cells grew normally in LB (Fig 3C) or MHB (Fig 3D). These data indicate that significant lysis did not occur in either medium over the course of 18 hours. D66 therefore rapidly disrupts bacterial membrane voltage without permeabilizing or lysing cells, and the bacteria can recover voltage and growth. A primary mechanism of D66 activity therefore appears to be depolarization, and, with time and/or higher

concentrations, energetic loss and/or compound accumulation permeabilize membranes without lysing cells.

## Genetic lesions in the *hns* gene correlate with resistance to D66 and reduced fitness in macrophages

We established whether strains resistant to D66 could be obtained first in a genetic background lacking *acrAB*, because deletion of this locus sensitized bacteria to the compound (Fig 2B and 2C), suggesting that D66 could be an AcrAB-TolC substrate and that selection for mutants in an *ΔacrAB* background would increase the probability of obtaining resistant mutants at other loci. We considered selecting for mutants in a Δ*tolC* background, but Δ*tolC* mutant strains are more severely attenuated than Δ*acrAB* strains [20,32], and it would be difficult to test recovered mutant strains for resistance to D66 in macrophages. Six independent isolates were evolved in an *ΔacrAB* background in the presence of increasing concentrations of D66, starting at 0.25x MIC and continuing stepwise until growth at 2x MIC was achieved over approximately eight passages. Analysis of whole-genome sequences revealed that all six resistant strains had acquired mutations in the dimerization domain of H-NS that were absent in vehicle-treated control strains [33,34] (Fig 4A). The *hns* mutations are predicted to diminish H-NS function [35], and loss-of-function mutations in *hns* enable the expression of efflux pumps, including *acrEF*, *acrD*, *mdtEF*, *macAB*, and *emrKY* [36,37], which could export D66. H-NS is required for *S*. Typhimurium virulence in mice [38], which correlates with replication within macrophages [39]. Consistent with these observations, the D66-resistant mutants survived poorly in macrophages, compared to the parent Δ*acrAB* mutant, which, as expected [20], accumulated to 60% of wild-type levels (Fig 4B). These results indicate that the resistant strains have decreased fitness during infection, prohibiting the testing of the mutants for D66 resistance in macrophages. Overall, the data show that H-NS contributes to D66 sensitivity, potentially by repressing efflux pumps.

## Selection for resistance mutants in sub-inhibitory concentrations of PMB with D66 yielded mutants with increased PMB resistance

We next established whether D66 resistant clones could be obtained in the presence of a subinhibitory concentration of PMB [0.4 ug/mL] and increasing concentrations of D66 up to 3x $MIC_{95}$. Six independent isolates were obtained. One of the isolates was set aside because resistance was not heritable. Five of the isolates were genetically resistant to the combination of PMB and D66. However, growth assays with the five resistant clones revealed significantly increased resistance to PMB (Fig 4C). The ease with which resistance to PMB was obtained is consistent with observations that multiple overlapping genetic pathways maintain outer membrane integrity and affect PMB resistance [40–43]. These data further confirm that a robust outer membrane normally protects bacterial cells from D66.

## D66 hyperpolarizes mitochondrial membranes but does not permeabilize host cell membranes

Mitochondrial membranes are similar in lipid composition to bacterial inner membranes in that they contain phosphatidylglycerol and cardiolipin [44] and may therefore be vulnerable to D66. To establish whether D66 alters the voltage of mitochondrial membranes in uninfected macrophages, we used the fluorescent dye tetramethyl rhodamine (TMRM). TMRM accumulates in the mitochondrial inner membrane and increases fluorescence in response to

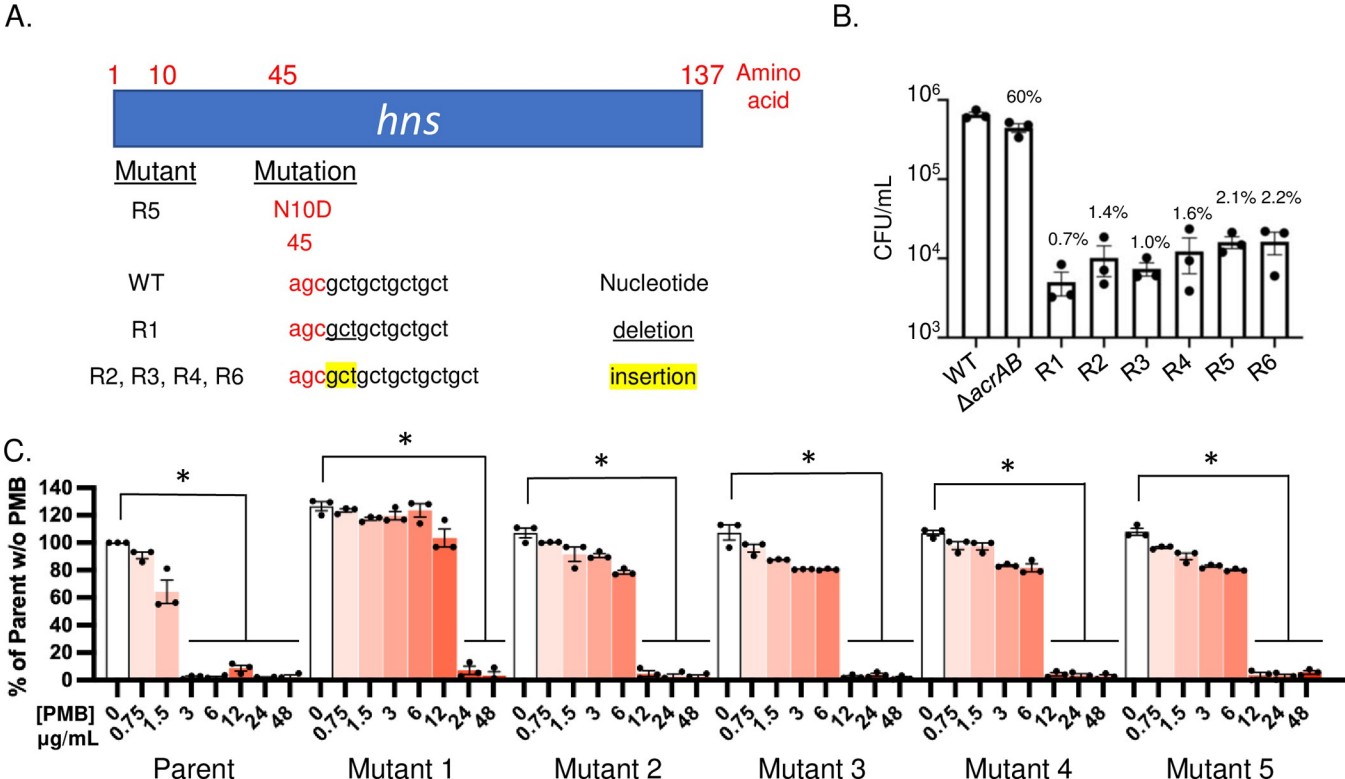

**Fig 4. Analysis of resistant mutants.** A, B), D66-resistant mutants selected for in an *ΔacrAB* mutant strain background. A) Diagram showing the *hns* mutation in six independent D66-resistant clones. B). RAW 264.7 cells were infected with *S.* Typhimurium SL1344, *ΔacrAB* or the six D66 resistant mutants as indicated. After 18 hours of infection, cells were lysed and plated for enumeration of CFU. Numbers above bars indicate percent of wildtype (SL1344) CFU/mL. Mean and SEM of biological triplicates with technical triplicates. A one-way ANOVA with a Dunnet's multiple comparison test: $P = 0.005$ for *ΔacrAB* and $< 0.001$ for all resistant mutants, compared to the wild-type strain. C) D66-resistant mutants selected for in the presence of PMB. The five independent clones identified were all resistant to PMB at concentrations 4-8X higher than the parent strain, which grew to an $OD_{600}$ of 1.2. A one-way ANOVA with a Dunnet's multiple comparison test: * $P < 0.05$ compared to the corresponding strain without PMB.

increased membrane potential [19,45]. RAW 264.7 cells were pre-loaded with TMRM and treated with DMSO, the protonophore carbonyl cyanide m-chlorophenyl hydrazone (CCCP), or D66. As anticipated, CCCP decreased TMRM fluorescence, reflecting membrane depolarization at concentrations effective in SAFIRE (7 µM) (Fig 5A). Treatment with D66 increased TMRM signal in a dose-dependent manner for the first two hours, suggesting membrane hyperpolarization, an indicator of cell stress [45–48]. Over time, samples treated with the highest concentration of D66 (56 µM) underwent a steady decline in signal, possibly reflecting compound aggregation and clearance. Mitochondria, therefore, appear to respond modestly to treatment of cells with D66.

To determine whether D66 permeabilizes mammalian cell membranes, we used a standard lactate dehydrogenase release assay (LDH) to monitor membrane leakage [49,50]. In uninfected RAW 264.7 cells, treatment with D66 had little effect on LDH release (Fig 5B). Exposure to pathogens radically changes the biology of mammalian cells [51], so we also measured LDH release in infected RAW 264.7 cells. D66 reduced the percentage of cells that released LDH in a dose-dependent manner (Fig 5B), consistent with the ability of the compound to reduce bacterial colonization and thereby improve macrophage viability. These data suggest that D66 is minimally toxic to mammalian cell membranes.

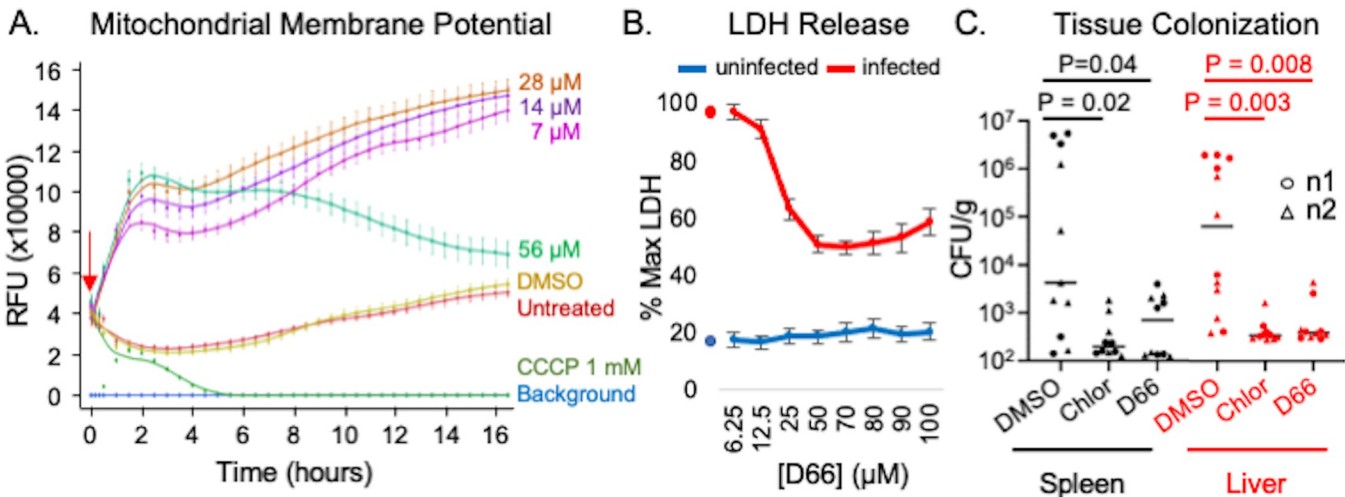

**Fig 5. D66 is well tolerated by eukaryotic cells and has antimicrobial activity in mice.** A) RAW 264.7 cells were incubated with the mitochondrial membrane potential indicator TMRM, treated (red arrow) with DMSO (0.5%), CCCP, or dilutions of D66, and imaged over time. Averages and SEM of three biological replicates with technical triplicates, normalized to time 0. B) RAW 264.7 cells that were uninfected or infected with *S.* Typhimurium SL1344 for 2 hours were treated with DMSO or D66 and monitored for LDH release after 16 hours. Averages and SEM of three biological replicates with technical duplicates, normalized to the maximum amount of LDH release (lysed cells; % Max LDH). Symbols on the Y-axis show the percentage of LDH released by DMSO- treated cells. C) C57Bl/6 mice were intraperitoneally inoculated with *S.* Typhimurium. At 10 minutes and 24 hours after infection, mice were dosed with 50 mg/kg of chloramphenicol or D66 by intraperitoneal injection. Mice were euthanized 48 hours after infection. The spleen and liver were homogenized and plated for enumeration of CFU. Significance was determined by Mann-Whitney.

## D66 reduces bacterial tissue colonization in mice

Since D66 had modest effects on host cells, we evaluated gross toxicity and pharmacokinetics in mice following a single dose. No adverse effects (hunching, tachypnea, or abnormal ambulation) were observed at an intraperitoneal dose of 50 mg/kg after 24 hours. The peak serum concentration observed was 3.5 μM, within range of the 6 μM that is effective in SAFIRE (Fig B in S1 Fig). The elimination half-life for D66 was estimated to be 3 hours with extensive distribution to tissues based on steady state volume of distribution ($V_{ss}$) of 43.3 L/kg. Under these conditions, D66 appears to be minimally toxic to mice and is present at levels compatible with testing for potency *in vivo*.

To establish whether D66 treatment affects *S.* Typhimurium colonization of tissues in mice, we inoculated C57Bl/6 mice intraperitoneally with $1 \times 10^4$ wild-type bacteria and then treated with 50 mg/kg of D66 intraperitoneally at 10 minutes and 24 hours post-inoculation [18,19,25]. All mice that received D66 survived in good condition out to 48 hours, at which time the spleen and liver were harvested. Enumeration of tissue CFU revealed that treatment with D66 reduced *S.* Typhimurium colonization in both tissues ($P < 0.05$, Mann-Whitney; Fig 5C). Thus, the compound was tolerated *in vivo* and had antibacterial potency.

## Discussion

### The effect of D66 on bacteria

Bacteria are assaulted by host soluble innate immune defenses in all body fluids, including in serum, the contents of phagolysosomes, and the cytosol [9–11]. Therefore, small molecules that are unable to breach the Gram-negative cell envelope in standard microbiological media may be able to gain access to bacteria during infection. These compounds could be identified by their ability to prevent bacterial survival during infection, as within the SAFIRE assay. D66 and JD1 appear to be examples of such molecules because they enable bacterial killing in

macrophages and in animals but only under broth conditions that compromise the outer membrane and/or efflux pumps. These observations further establish that Gram-negative bacteria are protected from the compounds by a combination of their outer membrane and efflux pumps. Both compounds inhibit *S*. Typhimurium growth in MIC assays, which utilize cells recovering from stationary phase at low cell density (OD 0.01). Both compounds also disrupt voltage across the bacterial inner membrane of mid-log phase cells (OD 0.4–0.6). However, only JD1 rapidly permeabilizes the inner membrane and kills bacteria: at 1x MIC, PI signal increases 33-fold within 30 minutes of JD1 treatment, compared to two-fold with D66. D66 therefore appears to have a more subtle effect, requiring higher concentrations and/or more time to permeabilize the inner membrane, and the bacterial cells recover over time from the damage wrought by D66. It is feasible that D66 is effectively diluted by a higher density of bacteria and/or that mid-log phase cells are more resistant to the compound than cells recovering from stationary phase. It is also possible that one or both compounds have unknown additional effects on the bacteria, the host cell, or both. Nevertheless, both D66 and JD1 appear to interact directly with bacterial cells in the context of cell envelope damage and to attack the inner membrane, suggesting that the inner membrane is vulnerable to small molecules during infection and that voltage disruption may be sufficient to augment bacterial killing by innate immune defenses and enable an intracellular pathogen to be eliminated by the host.

## Potentiation of D66 by PMB likely reflects PMB permeabilization of the outer membrane

In *Klebsiella pneumoniae*, AcrAB-TolC contributes to resistance to PMB, indicating that PMB is an AcrAB-TolC substrate [52]. If both D66 and PMB are exported by AcrAB-TolC, then potentiation of D66 by PMB could reflect competition for efflux. However, in *E. coli* and *S*. Typhimurium, PMB resistance is mediated primarily by LPS modifications [53,54], and likely at high PMB concentrations by the MdtEF-TolC efflux pump, which accumulates in *E. coli* upon PMB treatment [55]. Therefore, the simplest explanation for the observation that PMB potentiated D66 to inhibit bacterial growth is that PMB increases outer membrane permeability and D66 access to the bacterial cell.

## Bacteria appear to resist D66 based on a combination of outer membrane integrity and efflux pumps

Resistance to D66 in broth was selected for in an Δ*acrAB* mutant background and in the presence of PMB. The six independent D66 resistant clones recovered in a Δ*acrAB* background had predicted loss-of-function mutations in the gene encoding H-NS, which increases the expression of efflux pump genes, including *acrEF*, *acrD*, *mdtEF*, *macAB*, and *emrKY* [36,37]. The five independent clones recovered from selection in the presence of PMB and D66 had increased resistance to PMB. Thus, we were not successful at using a selection-for-resistance strategy to identify potential D66 target pathways. It follows that analysis of resistant mutants may not be the most efficient approach for gleaning mechanism of action for compounds that need help traversing the outer membrane and/or appear to be efflux pump exported [36]. Nonetheless, these observations reveal that bacteria normally protect themselves from D66 based on a combination of outer membrane integrity and export of the compound through efflux pumps.

## The mammalian host and D66

In cell culture infection experiments, D66 was active against *S*. Typhimurium in macrophages. However, for both bacterial strains, the compound had little effect on bacterial load in HeLa

cells for reasons that are not understood. We speculate that the microenvironment of the macrophage phagosome is more effective at permeabilizing the bacterial outer membrane and/or could modify the compound to increase potency.

D66 reduced viable bacteria in macrophages and in mice without obvious tolerability issues in uninfected or infected animals. However, uninfected macrophages do respond to D66 treatment, as revealed by mitochondrial hyperpolarization, an indicator of cell stress [45–48]. Mitochondrial membranes use multiple, complex compensation mechanisms to increase membrane voltage over time in response to depolarization [56]. For instance, hyperpolarization occurs upon treatment with agents that interfere with oxidative phosphorylation [56], ATP synthase [48], or proton consumption [57]. Therefore, hyperpolarization may not be due to a primary effect of D66 on this organelle. In addition, the lack of obvious murine pathology upon D66 exposure suggests that damage caused by D66 is minimal and/or that cells recover. Overall, the modest effects noted for this compound on mammalian cells and whole animals suggests there is value in exploring the use of compounds that target bacterial inner membranes as antibacterials.

## Conclusions

Basic science has the potential to identify new routes towards the development of antibacterials by suggesting unexplored bacterial structures, molecules, or processes as targets. Compounds that are distinct from existing antibacterials and have infection-dependent antimicrobial activity have been found using in-cell or in-host screens [58–61]. Follow-up studies to ascertain their general [19,30] and ultimately their molecular mechanisms of action will enable determination of whether such hit compounds could be developed into lead compounds. This approach requires the vision to study compounds that are not likely to become drugs but could suggest new directions for antibacterial research. JD1 and now D66 are examples of such compounds.

## Methods

### Ethics statement

This study was carried out in accordance with the recommendations in the *Guide for the Care and Use of Laboratory Animals* of the National Institutes of Health. Protocols were approved by the University of Colorado Institutional Committees for Biosafety and Animal Care (2445). Euthanasia method: carbon dioxide asphyxiation.

### Bacterial strains

*S.* Typhimurium (SL1344) [62], *S.* Typhimurium (14028s, ATCC), *S.* Typhimurium Δ*acrAB* (ALR1257) [20], *E. coli* (K-12 derivative BW25113 (wildtype) [63]), *E. coli* K-12 Δ*tolC* (JP313 delta tolC [64]; also called AD3644 and JLD1285).

### Media and reagents

Unless otherwise stated, bacteria were grown in LB at 37˚C with aeration. D66 is AW00798 from MolPort. To obtain mid-log phase cells, bacteria were grown overnight in LB, diluted the next morning 1:100 in fresh LB, and then grown to an $OD_{600}$ of 0.4–0.6). Cation-adjusted MHB was purchased from Sigma-Aldrich (90922).

## SAFIRE and CFU assays

SAFIRE assays were performed with RAW 264.7 (TIB-71) macrophages seeded at $5 \times 10^4$ in 100 µL of complete DMEM in 96-well tissue culture plates (Greiner, 655180) and incubated at 37˚C with 5% $CO_2$. *S.* Typhimurium (SL1344 with *sifB*::*gfp*) [65] was grown overnight in LB and diluted to $3 \times 10^7$ CFU/mL in complete DMEM. Twenty-four hours after seeding, 50 µL of bacterial cultures were added to each cell culture well, an approximate multiplicity of infection (MOI) of 30 bacteria to one RAW 264.7 cell. Plates were centrifuged at 500 *x g* for 2 minutes to synchronize the infection. Forty-five minutes after infection, 50 µL of DMEM containing 160 µg/mL gentamicin (Sigma-Aldrich) was added for a final gentamicin concentration of 40 µg/mL. At two hours after infection, cells were treated with vehicle (DMSO) or D66. At 17.5 hours after infection, PBS containing MitoTracker Red CMXRos (Life Technologies) was added to a final concentration of 100 nM. At 18 hours after infection, 16% paraformaldehyde was added to a final concentration of 4% and incubated at room temperature for 15 minutes. Cells were washed, stained with 1 µM DAPI and stored in 90% glycerol in PBS until imaging. After 16 hours of treatment, samples were imaged on a spinning disk confocal microscope, and a MATLAB algorithm calculated bacterial accumulation (GFP fluorescence) within macrophages, as defined by DAPI (DNA) and MitoTracker Red, a vital dye for mitochondrial voltage. GFP+ macrophage area is defined as the number of GFP-positive pixels per macrophage divided by the total number of pixels per macrophage, averaged across all macrophages in the field.

CFU assays were performed with RAW 264.7 cells seeded as above or with HeLa cells (ATCC CCL-2) seeded at $1 \times 10^4$ cells per 96-well. Cells were infected as described above with either *S.* Typhimurium SL1344 or 14028, as indicated at an approximate MOI of 30 bacteria per RAW 264.7 cell and 150 bacteria per HeLa cell. Plates were centrifuged and gentamicin treated as above. At 18 hours after infection, wells were washed twice with PBS and lysed with 30 µL 0.1% Triton X-100 in PBS for five minutes. Lysed cells were plated to L-agar and enumerated for CFU.

## Minimum inhibitory concentration determination

Overnight LB-grown cultures were diluted in LB or MHB to an optical density ($OD_{600}$) of 0.01 and distributed into polystyrene 96-well flat-bottom plates (Greiner, 655185). D66 was added at concentrations up to 150 µM, near the limit of solubility. The final DMSO concentration was at or below 2%. Where indicated, PMB [0.5 µg/mL] was added prior to D66. Plates were grown at 37˚C with shaking and $OD_{600}$ was monitored (BioTek Synergy H1 or BioTek Eon). MICs were defined as the concentration at which 95% of growth was inhibited ($OD_{600}$).

## Novobiocin potentiation assays

Overnight LB-grown cultures were diluted in LB to an optical density at 600 nm ($OD_{600}$) of 0.01 and distributed into polystyrene 96-well flat-bottom plates (Greiner, 655185). D66 was added at concentrations up to 150 µM, near the limit of solubility, and novobiocin was added up to a concentration of 100 µg/mL. The final DMSO concentration was at or below 2%. Plates were grown at 37˚C with shaking for 18 hours and $OD_{600}$ was monitored (BioTek Synergy H1).

## Bacterial membrane potential assays

Membrane potential was measured using the potentiometric fluorescent probe $DiSC_3(5)$ (Invitrogen). Mid- log phase cells were diluted to an $OD_{600}$ of 0.4. $DiSC_3(5)$ was added to a final

concentration of 2 μM and the culture was incubated at 37°C in a rotator for 15 minutes. Cells were captured on a 0.45 μm Metricel membrane filter (Pall), resuspended in fresh LB with 0.5 μg/mL PMB (to enable $DiSC_3(5)$ and D66 to traverse the outer membrane), and distributed (200 μL) into black polystyrene 96-well plates (Greiner, 655076). Plates were monitored (ex650/em680 nm) on a BioTek Synergy H1 plate reader. After baseline fluorescence was recorded, compound was added to the desired final concentration and measurements were recorded for an additional 30 minutes. This assay was not performed in the BW25113 Δ*tolC* strain given the difficulty of loading $DiSC_3(5)$ into the inner membrane of the this strain [19].

### Propidium iodide membrane barrier assays

Compound, DMSO, or SDS was added to mid-log phase cells to the desired concentration, and cultures were sampled at 0, 10, 15, 20, 30 and 45 minutes. Five minutes before harvesting, PI [10 μg/mL] (Life Technologies) was added. Cells were pelleted, washed twice, resuspended in PBS, and monitored (ex535/em617 nm) using a BioTek Synergy H1 plate reader.

### Growth curves and kill curves

Mid-log phase cultures were sampled at time 0 and then compound or vehicle control (DMSO) was added. Cultures were incubated at 37°C with agitation. At the time intervals indicated, aliquots were monitored for $OD_{600}$ and plated for CFU enumeration. Data for $OD_{600}$ and CFU/mL were normalized to time 0.

### Evolution of resistant mutants and genetic analysis

To ensure that all isolates started with the same genetic background, a single colony of wild-type S. Typhimurium was resuspended and then distributed into six independent M9 low magnesium broth cultures containing 0.25x MIC of D66. Each day growth was visible, cultures were diluted 1:100 into fresh medium containing an additional 0.25x MIC D66 until growth at 2x MIC was achieved (~8 passages). Isolates were recovered on LB agar and tested for heritable resistance with 2x MIC D66. Genomic DNA from overnight cultures of resistant mutants and two solvent-treated controls from the same single colony were extracted with the E.Z.N.A bacterial DNA kit (Omega Bio-tek). Library preparation (Nextera XT) and sequencing (MiSeq V2 2x150 paired end) was performed by the BioFrontiers Sequencing Facility at the University of Colorado Boulder. Data were analyzed for mutations using Snippy (https://github.com/tseemann/snippy).

### Mitochondrial membrane determination with TMRM

Experiments were performed with RAW 264.7 cells between passages one and six. Cells were grown in complete DMEM to a confluency of 70–90%. Cells were scraped, washed, resuspended and diluted in complete DMEM to a final concentration of $5x10^5$ cells/mL. Cells (100 uL) were transferred to a 96-well glass bottom plate (0.17mm, Brooks Life Sciences) and incubated for 23.5 hours at 37° C with 5% $CO_2$. The medium was exchanged for 100 uL of complete FluoroBrite DMEM with TMRM [100 nM] and incubated for 30 minutes. The medium was exchanged for 150 uL of complete FluoroBrite DMEM. Cells were imaged on a Yokogawa CellVoyager CV1000 Confocal Scanner System with a 20x/0.75NA objective and an environmentally controlled multi-well chamber over 30 minutes with images acquired every 10 minutes. Compounds were added (50 uL) with a multichannel pipet to obtain the desired concentration and a final volume of 200 uL with 0.5% DMSO. Cells were imaged over 16 hours with acquisition every 30 minutes of two fields of view per well. Five images over a z-

dimension of 15 μM were sampled per field. The resulting volumes were converted into maximum intensity projections and TMRM foreground signal was extracted via a MATLAB R2018a (MathWorks) script and normalized to time zero for each field.

## LDH assays

An LDH-cytotoxicity assay kit (Abcam ab65393) was used according to the manufacturer's instructions. RAW 264.7 cells were seeded and remained uninfected or with infected as described for the CFU assay with SL1344.

## Murine pharmacokinetic analyses and infections

Three female C57BL/6 mice were injected intraperitoneally (IP) with 50 mg/kg of D66 formulated in DMSO (50 μL). This dose was selected based on D66 solubility, which suggested the compound distributes evenly within the mouse such that a dose of 54 μg per 20 g mouse is needed to achieve a concentration of approximately 7.8 μM, the $IC_{50}$ of D66 against *S*. Typhimurium in macrophages. Three mice were initially dosed and observed for 24 hours to determine tolerability. Following the lack of observable toxicity and gross pathological lesions in liver, kidney, and gastrointestinal tissues, another 12 mice were treated with the same dose of D66 and plasma samples collected at 0.5,1, 4, and 8 hours by cardiac exsanguination under isoflurane anesthesia. Plasma D66 levels were measured using a liquid chromatography coupled to tandem mass spectrometry (LC/MS/MS) assay by the University of Colorado Cancer Center Drug Discovery and Development Shared Resource. The assay used monitored the transition of D66 (376 *m/z* → 176 *m/z*) and was linear from 1–1000 ng/ml with an accuracy and precision of 90.3% ± 10.4% (%CV) based on quality control (QC) samples included with analyzed unknown samples. The peak serum concentration observed was 3.5 μM, and the terminal half-life was 3.0 hours.

Female C57Bl/6 7–8-week-old mice were IP inoculated with *S*. Typhimurium strain SL1344. Six mice per cohort were IP-treated with 100 μL of vehicle (50% DMSO), 50 mg/kg of chloramphenicol, or 50 mg/kg of D66 at 10 minutes and 24 hours post-infection. Mice were euthanized at 48 hours by $CO_2$ asphyxiation, followed by cervical dislocation [7]. Spleen and liver were collected, homogenized in 1 mL PBS and serially diluted for plating to enumerate CFU. The experiment was performed twice independently with $7 \times 10^3$ CFU, $3 \times 10^4$ CFU respectively, as determined by plating. A ROUT test for outliers and a Mann-Whitney test for significance were performed in GraphPad Prism.

## Supporting information

**S1 Fig. Fig A. D66 quenches the fluorescent dye $DiSC_3(5)$ in a concentration-dependent manner.** A) Control wells (without bacterial cells) containing medium with 2 mM $DiSC_3(5)$ and DMSO or compound, as indicated, added at time 0. B) Data from Fig 3A normalized to DMSO at time 0 but without correction for the quenching effect of D66 observed in panel A. JD1 was included as a control. **Fig B. D66 pharmacokinetic parameters**. Values were calculated by compartmental modeling using Phoenix WinNonlin. Data fit a two-compartment model (r = 0.9976) with bolus dosing. A) Decay curve. B) Parameter values.
(PDF)

## Acknowledgments

We thank all the members of the Detweiler laboratory for insightful discussions and technical help. We are grateful to C.A. Ewing, P. M. Roeder, T. Sammakia, and J.A. Villanueva for

comments on the manuscript. We thank the MCDB Light Microscopy Facility at the University of Colorado Boulder. We also thank the University of Colorado BioFrontiers Institute Next-Gen Sequencing Core Facility for Illumina sequencing and library construction.

## Author Contributions

**Conceptualization:** Jamie L. Dombach, Joaquin LJ Quintana, Corrella S. Detweiler.

**Data curation:** Jamie L. Dombach, Joaquin LJ Quintana, Toni A. Nagy, Daniel L. Gustafson, Corrella S. Detweiler.

**Formal analysis:** Jamie L. Dombach, Joaquin LJ Quintana, Samual C. Allgood, Toni A. Nagy, Daniel L. Gustafson, Corrella S. Detweiler.

**Funding acquisition:** Corrella S. Detweiler.

**Investigation:** Jamie L. Dombach, Joaquin LJ Quintana, Samual C. Allgood, Toni A. Nagy, Daniel L. Gustafson, Corrella S. Detweiler.

**Methodology:** Jamie L. Dombach, Joaquin LJ Quintana, Daniel L. Gustafson, Corrella S. Detweiler.

**Project administration:** Jamie L. Dombach, Corrella S. Detweiler.

**Resources:** Corrella S. Detweiler.

**Software:** Joaquin LJ Quintana.

**Supervision:** Jamie L. Dombach, Corrella S. Detweiler.

**Validation:** Jamie L. Dombach, Joaquin LJ Quintana, Samual C. Allgood, Toni A. Nagy, Daniel L. Gustafson, Corrella S. Detweiler.

**Visualization:** Jamie L. Dombach, Joaquin LJ Quintana, Samual C. Allgood, Toni A. Nagy, Daniel L. Gustafson, Corrella S. Detweiler.

**Writing – original draft:** Jamie L. Dombach, Joaquin LJ Quintana, Corrella S. Detweiler.

**Writing – review & editing:** Jamie L. Dombach, Joaquin LJ Quintana, Toni A. Nagy, Daniel L. Gustafson, Corrella S. Detweiler.

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
