## [Decision Letter · Decision Letter 0]

17 Mar 2022

Dear Corrie,

Thank you very much for submitting your manuscript "A small molecule that disrupts S. Typhimurium membrane voltage without cell lysis reduces bacterial colonization of mice" for consideration at PLOS Pathogens. As with all papers reviewed by the journal, your manuscript was reviewed by members of the editorial board and by several independent reviewers. In light of the reviews (below this email), we would like to invite the resubmission of a revised version that takes into account the reviewers' comments. We encourage you to experimentally address the lack of PMB as a control in Fig2 A-D as well as the evolution of resistance in the WT strain exposed to sub-lethal concentrations of PMB. 

We cannot make any decision about publication until we have seen the revised manuscript and your response to the reviewers' comments. Your revised manuscript is also likely to be sent to reviewers for further evaluation.

Sincerely,

Sophie Helaine

Associate Editor

PLOS Pathogens

Renée Tsolis

Section Editor

PLOS Pathogens

Kasturi Haldar

Editor-in-Chief

PLOS Pathogens

orcid.org/0000-0001-5065-158X

Michael Malim

Editor-in-Chief

PLOS Pathogens

orcid.org/0000-0002-7699-2064

Reviewer's Responses to Questions

**Part I - Summary**

Reviewer #1: The manuscript by Dombach et al. describes the identification and testing of a small molecule that reduces Salmonella CFU within macrophages and in a mouse model of infection. Attempts to determine the mechanism of action have led them to the bacterial inner membrane, where it disturbs the membrane voltage without permeabilization of the membrane. Mutants with increased resistance to the compound were identified, which mapped to H-NS, but it is not likely the target. Treatment of mammalian cells also showed an effect of the compound on the mitochondrial membrane, where it resulted in hyperpolarization. The identification of a small molecule that affects Salmonella membranes is of interest in the antimicrobial field. However, there are some issues with the data that is presented and the findings reported move this work forward, but not in a significant way. More data is needed.

Reviewer #2: Dombach et al. report D66 as a new small molecule that prevents the survival of Gram-negative pathogen in macrophages. In their manuscript, the authors show that D66 inhibits bacterial growth under specific conditions or in specific genetic backgrounds. They also found that mutations in hns gene confer resistance to D66 in vitro. Finally, they show that D66 can reduce bacterial tissue colonization in mice. The manuscript is original, timely and could interest a large audience. The findings are interesting but additional work is essential to clarify the current model. These experiments could severely change the interpretation of the manuscript and the current discussion.

Reviewer #3: The authors use a screening platform based on tissue culture cell infection assay to identify compounds that specifically target S. Typhinurium in macrophages. Here the authors charachterize D66, a small molecule with anti S. Typhinurium activity in macrophages. They find that macrophage mediated damage ot the cell envelope facilitates D66 antimicrobial activity, but D66 itself is not an outer membrane permeabilizing agent. Instead, once D66 is capable of getting through the OM, it then perturbs the voltage across the cytoplasmic membrane without permeabilizing it. Despite hyperpolarization of mitochondrial membranes in response to D66, infected macrophages survived better with D66 treatment, presumably down to the anti-S. Typhinurium effects. Furthermore, D66 showed efficacy in a mouse model of infection, without associated toxicity. Overall, this is a well thought out study. The data is clear and appropriately interpreted.

**Part II – Major Issues: Key Experiments Required for Acceptance**

Reviewer #1: 1. The primary message of this manuscript, that the cytoplasmic membrane is a good antimicrobial target, was the same message as these authors’ previous PLoS Pathogens manuscript on JD1.

2. Was D66 identified in the screen in which they identified JD1? This is not clear from the manuscript. Why was this compound chosen for study?

3. The authors should discuss why they think the compound works in macrophages but not epithelial cells.

4. The authors should focus just on D66. They occasionally bring a previously studied compound JD1 into the discussion. This seems a bit distracting at times.

5. Figure 1D. It is unclear why there are low IC50 ratios described when this data told the authors that there is no effect of the compound on HeLa cells.

6. Figure 2A and B. Why didn’t the authors do a typical growth curve, and instead use stationary phase bacteria? A typical growth curve examining D66 effects should be performed like in 3C- F. Also, it is unclear why there are the blank spaces to the left of the data in each panel. Finally, why didn’t the authors use polymyxin alone as a control? This needs to be included as you cannot just rely on the literature source for the concentration that is sub-lethal.

7. Figure 2C and D. The checkerboard assays are unclear. It appears there is only activity at the highest concentration of novobiocin. There are some sporadic changing of the colors in panel C. Why is this? The authors should try to perform a fractional inhibitory concentration index from the checkerboard data. This can identify synergy, additivity, etc.. The authors stated that there was synergy online 170, but there is no data affirm that. Finally, a control of a compound that does affect the outer membrane (like polymyxin B) would have been useful as a control in this figure.

8. In the in vitro evolution experiments, the authors should have used two or more independent colonies, as just performing the evolution with one colony can lead to siblings from an early mutational event. In fact, you did have four of the sequenced mutants that were identical. Also, for figure 4 and this data, the authors should determine the sensitivity of the evolved strains to D66 in comparison to the parent strain. Finally, putting the H-NS mutations into a tolC mutant background and the WT strain is warranted to determine if D66 resistance mutations in H-NS are only effective in the acrAB mutant background.

9. Figure 5. In panel A, the authors state that the compound D66 has a dose dependent hyperpolarization effect. That is not true from the graph as the 56 umolar concentration shows much less polarization than the lower concentrations, which escalate with increasing dose.

10. Figure 5. Examination of infected cells in the LDH assay complicates the scenario as infection causes LDH release. It looks clear from the uninfected that there is no LDH release. I’d just present that.

11. In the animal experiments, the PK data was achieved with one dose of D66 but the effect on Salmonella infected mice was performed with two doses of D66. These are not comparable. Also, in line 320, the authors’ state that the compound was tolerated in vivo but that is only over a 48 hour period after which they sacrifice the mice. This should be stated with the caveat that mice were only examined after 2 days, where pathology might not even have a chance to develop.

12. Did the authors consider using efflux pump poisoning agents to see if that increased the activity of their compounds?

Reviewer #2: -Line 132 - “D66 inhibits bacterial growth under conditions that compromise the cell envelope”. The data of this section suggest that D66 is pumped out by AcrAB efflux pump (as suggested by the authors on line 231). Stresses on the cell envelope could affect AcrAB activity. Authors should test the activity of AcrAB in presence of PMB (described here: PMID: 27381291; Hoecth or PI assay). In addition, it is well known that PMB is a substrate of the AcrAB efflux machinery in enterobacteria. Therefore, PMB could sensitize cells to D66 by directly competing for the same efflux machinery. Finally, it is unclear if it is D66 or the PMB itself that affects bacterial physiology in these conditions. The authors should test the effect of PMB on the acrAB mutant, as done on panels 2A and B.

-The importance of AcrAB suggests that efflux pumps limit the entry of the D66 inside the cells. Since D66 is presented as an alternative to traditional antibiotics, authors should check if mutations in AcrAB that confers antibiotic resistance (e.g. fluoroquinolones) protect Salmonella against the D66 compound during infection.

- Line 176 – In this section, the authors used PI to assess membrane permeabilization during D66 treatment. However, since PI labeling is also dependent on the activity of AcrAB, these results could be misinterpreted. One hypothesis could be that D66 promotes PI labeling by competing for the same efflux machinery. Please clarify this section.

- According to their data, the simplest model is that D66 is directly targeting Hns. Did the authors test this hypothesis? Mutations in hns could limit the interaction between the drug and Hns itself. These data are extremely interesting but are unclear and not sufficiently addressed in the manuscript. The authors suggest in their discussion that loss of Hns could promote the expression of alternative efflux pumps. The authors should perform efflux assay (with PI or Hoecht) with their Hns mutants to validate their hypothesis.

Reviewer #3: The search for for resistant mutants is performed in a mutant that already has some compromised fitness, in order to create in vitro sensitivty. The mutations selected for in this mutant background may be synthetically lethal to the bacteria in macrophages while easily maintained in a wt background in macrophages. The selection of resistant mutants should be performed in the presence of sub-inhibitory PMB as in Figure 2. That way, you generate resistance directly in the envelope compromising conditions, similar to those experienced in macrophages and then test that mutant for fitness defects in macs. It would be a far cleaner way to determine if resistant mutants, retaining fitness, are quickly selected for by D66 treatment.

**Part III – Minor Issues: Editorial and Data Presentation Modifications**

Reviewer #1: 13. Line 54. “..relatively impervious to treatment” is too broad of a statement.

14. Line 57 to 58. This sentence is out of place in the author summary.

15. Line 72. Perhaps change “in the periplasm” to “from the periplasm”

16. Line 75. Change the sentence to “…(e.g. serum) and the contents of phagolysosome after phagocytosis”.

17. Line 78. Move “during infection” to the front of the sentence and remove the word “thus”

18. Line 293- 294. There are many other differences between bacterial and mammalian cell membranes than charge.

19. Line 410 through 411. The D66 statement is not a sentence.

20. Line 424. Please fix the spelling of gentamicin

Reviewer #2: - Line 67 – “The negative charges on LPS are stabilized by recruited cations, creating a coat around the bacterium that excludes many antibiotics”. This sentence is a bit counterintuitive since antibiotics are effective against bacteria. This sentence suggests that LPS allows bacteria to grow in presence of antibiotics by excluding them (antibiotic resistance). Please clarify.

- Table1 - Define IC50, MIC50, and cMIC95 in the legend of the table

- Line 166 – The rationale used by the authors in this section is unclear. The sensitivity of the acrAB mutant suggests that D66 needs to enter the cell to mediate its effect. Therefore, the study of the outer membrane seems counterintuitive. Please clarify this point in this section and in the discussion.

Reviewer #3: Although the data support D66 not damaging the outer membrane, it would make more sense to include polymyxin B nonapeptide as a positive control in the novobiocin synergy assay in Figure 2. Here, you expect a 100-fold increase in sensitivity to novobiocin, which would confirm the appropriateness of the experimental procedures in ruling in, or out, OM damage.

PLOS authors have the option to publish the peer review history of their article (what does this mean?). If published, this will include your full peer review and any attached files.

Reviewer #1: No

Reviewer #2: No

Reviewer #3: No
---

## [Editor Report · Decision Letter 1]

19 May 2022

Dear Corrie,

We are pleased to inform you that your manuscript 'A small molecule that disrupts S. Typhimurium membrane voltage without cell lysis reduces bacterial colonization of mice' has been provisionally accepted for publication in PLOS Pathogens.

Best regards,

Sophie Helaine

Associate Editor

PLOS Pathogens

Renée Tsolis

Section Editor

PLOS Pathogens

Kasturi Haldar

Editor-in-Chief

PLOS Pathogens

orcid.org/0000-0001-5065-158X

Michael Malim

Editor-in-Chief

PLOS Pathogens

orcid.org/0000-0002-7699-2064
---

## [Editor Report · Acceptance letter]

7 Jun 2022

Dear Dr. Detweiler,

We are delighted to inform you that your manuscript, "A small molecule that disrupts * S.* Typhimurium membrane voltage without cell lysis reduces bacterial colonization of mice," has been formally accepted for publication in PLOS Pathogens.

Best regards,

Kasturi Haldar

Editor-in-Chief

PLOS Pathogens

orcid.org/0000-0001-5065-158X

Michael Malim

Editor-in-Chief

PLOS Pathogens

orcid.org/0000-0002-7699-2064